# Incidence of Drug-Resistant Enterobacteriaceae Strains in Organic and Conventional Watermelons Grown in Tennessee

**DOI:** 10.3390/foods11213316

**Published:** 2022-10-22

**Authors:** Onyekachukwu Akaeze, Agnes Kilonzo-Nthenge, Dilip Nandwani, Abdullah Ibn Mafiz, Maureen Nzomo, Tobenna Aniume

**Affiliations:** 1Department of Agriculture and Environmental Sciences, Tennessee State University, 3500 John A. Merritt Boulevard, Nashville, TN 37209, USA; 2Department of Human Sciences, Tennessee State University, 3500 John A. Merritt Boulevard, Nashville, TN 37209, USA

**Keywords:** Enterobacteriaceae, antimicrobial resistance, organic watermelon, conventional watermelon

## Abstract

The production and consumption of organic fresh produce have constantly increased since the 1990s. Consumers prefer organic produce because it does not contain synthetic chemical residues that are often implicated in health problems. The contamination of fresh produce by pathogenic *Enterobacteriaceae* strains remains a major challenge, and is responsible for frequent foodborne disease outbreaks. The use of antibiotics has proved an effective treatment, but the increase in occurrences of antibiotic resistance is becoming a health challenge. This study seeks to establish the presence of antimicrobial resistance in *Enterobacteriaceae* on organic and conventional watermelon fruits. Watermelons used for this study were cultivated at the Tennessee State University Certified Organic Farm, Nashville. At harvest, nine fruits were selected from among fruits lying on plastic mulch, and nine from fruits lying on the soil of both organic and conventional plots. These were placed in sterile sample bags for microbial analysis. Spread plating technique, API 20E, and apiweb software were used for microbial isolation and identification. Identified strains were tested for antimicrobial resistance against 12 common antibiotics. Seventeen Enterobacteriaceae strains were isolated and identified. Isolates were susceptible to gentamycin, ciprofloxacin, and chloramphenicol, but were resistant to cefoxitin. *Citrobacter freundii* showed a 14.3% resistance to Streptomycin. *Pantoea* spp. and *Providencia rettigeri* showed 50% and 100% resistance to tetracycline. Findings from this study confirm the presence of antibiotic-resistant Enterobacteriaceae strains on organic watermelons in Nashville, TN.

## 1. Introduction

Organic vegetable production and consumption have increased significantly during the past 20 years [1]. Consumption of fresh, nutrient-rich organic fruits and vegetables like watermelon contributes to a more productive and healthier lifestyle [1,2,3]. There has been an increase in cases where people become sick and sometimes hospitalized after consuming fresh produce. Common causes of foodborne disease are contamination by bacteria, viruses, and parasites or their toxins. Bacteria of the Enterobacteriaceae family are the most common human and opportunistic pathogens [4]. Their presence above threshold levels has frequently been associated with enteric infections [5,6]. In the United States, about 31 major pathogens (including *Salmonella* spp., *Clostridium perfringens*, *Campylobacter* spp., *Toxoplasma gondii*, and *Listeria monocytogenes*) are implicated in over 9.4 million annual cases of foodborne diseases. Symptoms include nausea, vomiting, stomach cramps, and diarrhea. The severity may range from mild (lasting for a few hours or days) to severe (resulting in hospitalization and sometimes death) depending on the presence of underlying health conditions, a weak or compromised immune system, age, and overall health status of the individual at the time of infection.

Members of the Enterobacteriaceae family responsible for foodborne diseases, such as *Klebsiella* spp., *Proteus* spp., *E. coli*, and *Salmonella,* are found not only in human and animal gut microbiota but also in the soil [1,7]. Agricultural soils under organic management systems have been reported to have a significantly higher microbial diversity and count [8] when compared to conventional agricultural soils. While a rich soil microbial diversity is an indicator of good soil health, it also poses food safety concerns, as agricultural soil, irrigation water, manure, or grazing animals may be sources of contamination for fresh produce. Furthermore, the consumption of crops such as watermelons and other fresh produce whose edible parts are in direct contact with the soil pose a higher risk of infection [9].

The first and often most effective method of treatment for foodborne diseases is the use of antibiotics [10,11]. Antibiotics have played a major role in significant advancements in medicine and surgery [11], resulting in the saving of many lives. However, many microbial species are becoming resistant to antibiotics [4,12], resulting in the reduced efficacy of antibacterial, antiparasitic, antiviral, and antifungal drugs [11]. Other implications of microbial resistance to antibiotics include delays in providing the right treatment to patients, health complications, and death. Additionally, a patient may need more care as well as alternative and more expensive antibiotics, which may have more severe side effects. Although antibiotic resistance is a natural occurrence resulting from gene mutation, selective pressure from excessive and inappropriate use of antibiotics has been reported to increase the emergence and spread of antibiotic-resistant bacteria. Identification of antibiotic resistance is key in the development of new antibiotic and management of foodborne diseases. The objectives of this study are as follows. First, to screen for antimicrobial resistance in Enterobacteriaceae strains isolated from watermelons grown in the Tennessee State University farm under organic and conventional management systems. Second, to determine the effect of plastic mulch on microbial load on watermelon under organic and conventional management systems.

## 2. Materials and Methods

### 2.1. Experimental Location and Design

The experiment was conducted at the Certified Organic Farm of Tennessee State University, Nashville TN, during the Summer of 2020. Organic and conventional Crimson Sweet watermelon were sown in the greenhouse for four weeks under routine nursery practices. Healthy seedlings were then transplanted to the field. First, the land was prepared by clearing and then plowed. Next, 5 ft wide black plastic mulch (Berry hill irrigation, Buffalo Junction, VA, USA) and drip tape (for irrigation) were installed on raised beds 7 ft apart. The field experiment was laid in a randomized complete block design (RCBD) and replicated three times. At the onset of flowering, fertilized flowers were carefully trained to sit on the plastic mulch, while others were in direct contact with the soil. At harvest, 9 fruits were selected from among fruits lying on plastic mulch, and 9 from fruits lying on the soil of both organic and conventional plots. The fruits were then transported to the laboratory in sterile sample bags for microbial analysis.

### 2.2. Microbial Isolation

In the laboratory, each selected watermelon was thoroughly scrubbed in 20 mL of peptone water to dislodge the microbes present on fruit surfaces. Next, seven-fold serial dilutions were prepared in tryptic soy broth (TSB) in four replicates for microbial analysis and quantification. Microbial isolation and purification were performed by the spread plate technique on violet red bile agar (Oxoid, Basingstoke, Hants, UK) and incubated at 37 °C for 24 h. Colonies with characteristic red to dark purple with red-purple halos were tentatively presumed to be of the *Enterobacteriaceae* and were counted. A more precise identification was further done by transferring pure colonies to tryptic soy agar (TSA) which were then incubated for 24 h at 37 °C.

### 2.3. Microbial Identification

After incubation, colonies were biochemically identified by using the API 20E (bioMerieux, Hazelwood, MO, USA) test methods. Strips were inoculated with samples following the manufacturer’s instructions, and the isolates were identified by using the apiweb software (bioMerieux SA 1.4.1-3 version). Enterobacteriaceae identified at and above the 90% confidence level were recorded in this study. Identified isolates were stored at −80 °C in 50% glycerol for antimicrobial susceptibility testing.

### 2.4. Antimicrobial Susceptibility Test

The antimicrobial susceptibilities of all identified isolates were determined following the methods of [4,13]. TSB cultures were prepared and allowed to stand overnight at 37 °C, after which they were spread evenly on Mueller–Hinton agar plates (Difco, BD) using a sterile cotton-tipped applicator (Puritan) and allowed to stand for 10 min at ambient temperatures. Antibiotic susceptibility disks were applied on Mueller–Hinton plates using Oxoid disc dispenser, and then incubated at 37 °C for 24 h, after which plates were observed for inhibition zones.

Twelve antibiotic susceptibility disks, listed here with the disk strength in parentheses, were used: ciprofloxacin (CIP; 5 mcg), ceftriaxone (CRO; 30 mcg), cefoxitin (FOX; 30 mcg), tetracycline (TE; 30 mcg), gentamicin (CN; 10 mcg), chloramphenicol (C; 30 mcg), streptomycin (S; 10 mcg), neomycin (N; 30 mcg), cefpodoxime (CPD; 10 mcg), nalidixic acid (NA; 30 mcg), cephalothin (KF; 30 mcg), and kanamycin (K; 30 mcg). Results were interpreted as resistant or susceptible based on the Clinical and Laboratory Standards Institute recommendations [14]. *Escherichia coli* ATCC 25922 and *Staphylococcus aureus* ATCC 25923 were used as control strains. Reference standard bacterial strains were verified simultaneously with controls.

### 2.5. Statistical Analysis

All plate count data were converted to log CFU per ml values before statistical analysis. The antibiotic resistance values are expressed as percentages. Statistical analysis was conducted using the SAS statistical software (version 9.4, SAS Institute, Cary, NC, USA).

## 3. Results and Discussion

### 3.1. Enterobacteriaceae

In this study, 17 Enterobacteriaceae strains were identified in organic and conventional watermelon fields in Nashville, TN (Table 1). Enterobacteriaceae strains have been reported to cause many foodborne diseases [4], hence, the presence and number of *Enterobacteriaceae* strains have been used as reliable food safety indicators [15]. In our study, *Enterobacter cloacae* had the highest frequency of occurrence (14.04%), closely followed by *Citrobacter freundii* (12.28%), *Aeromonas hydrophila, Chryseomonas luteola,* and *Klebsiella oxytoca* (each 7.02%). *Enterobacter sakazakii, Klebsiella Pneumonia, Providencia rettigeri,* and *Pseudomonas flourescence* had the least occurrence, each at 1.75%. *Citrobacter freundii* strain occurred on the surfaces of watermelons on plastic mulch both on organic and conventional fields. *Enterobacter aerogenes, Providential alcalifactiens,* and *Klebsiella oxytoca* strains occurred on the surfaces of watermelons in direct contact with the soil in both organic and conventional fields. The Enterobacteriaceae strains isolated in this study were similar to those reported in a similar study [16], where *Citrobacter* sp. *Klebsiella* sp., *Enterobacter* sp. *Proteus vulgaris,* and *Pseudomonas* sp. were isolated from watermelon. Some strains isolated and identified in this study, such as *Klebsiella pneumonia* and *Klebsiella oxytoca,* have previously been isolated from watermelon, lettuce, spinach, cilantro, tomatoes, and strawberries, and have been implicated in severe infections [4,17], thus proving that fresh fruits and vegetables may be vehicles for foodborne illnesses. Similar studies, such as [18] describe *Klebsiella* sp. as opportunistic pathogens that can lead to severe diseases, such as septicemia, pneumonia, and urinary tract and soft tissue infections. Additionally, *Enterobacter sakazakii,* which has been isolated from fresh-cut apple, cantaloupe, strawberry, watermelon, cabbage, carrot, cucumber, lettuce, and other fresh produce, has been identified as an emerging foodborne pathogen that has caused illnesses and deaths in infants and elderly immunocompromised adults [19].

### 3.2. Effects of Farming System and Plastic Mulch on Enterobacteriaceae Count

While there is numerous literature on the effect of farming systems on the microbial dynamics of many vegetables, there is a dearth of literature that focuses on the effects of farming systems on watermelon. Results from this study focus on the effects of organic and conventional farming systems on Enterobacteriaceae count in watermelon. Figure 1 shows that organic and conventional farming systems had a significant effect on the microbial count. Watermelon grown organically had a higher microbial count of 9.4013 log CFU, while that grown conventionally had a lower microbial count of 9.3532 CFU at *p* ˂ 0.05. Figure 2 shows that watermelon fruits in direct contact with the ground had a significantly higher microbial count of 9.4952 log CFU when compared to those on plastic mulch (9.2900 log CFU). Higher microbial diversity was observed in organic watermelon (eleven different strains on the ground and nine on plastic mulch) when compared to conventional (four on the ground and two on plastic mulch) watermelon. Furthermore, a higher microbial count of Enterobacteriaceae strains was also observed on watermelon directly on the ground when compared to those on plastic mulch. In their study, [20] evaluated the effects of organic and conventional production systems on microbiological analyses of lettuce, radish, carrot, and beetroot. The results showed that vegetables cultivated under an organic management system had a significantly higher load of Enterobacteriaceae compared to those conventionally cultivated. Similar findings were also reported by other researchers such as [21,22]. This may be because organic agriculture focuses on the use of natural processes to improve soil microbial activities [23]. Plastic mulch as a barrier on the soil surface may reduce the number of microbes reaching the surfaces of fruits, thereby resulting in a lower microbial count and diversity while also causing changes in soil physical and chemical properties and microbial activities [24]. This is in agreement with the findings in our study. Watermelon fruits on plastic mulch had a significantly lower microbial count compared to watermelon fruits in direct contact with the ground.

### 3.3. Susceptibility Testing

Antimicrobial resistance in Enterobacteriaceae strains is a major food safety concern, especially in fresh produce. Although organic products are regarded by consumers as safe for consumption due to the absence of deleterious chemical residues, findings from this study show that the presence of disease-resistant Enterobacteriaceae strains on organic watermelons may pose food safety concerns. Table 2 shows the response of 17 Enterobacteriaceae strains to 12 common antibiotics. Enterobacteriaceae showed varied sensitivity to the antibiotics evaluated. Furthermore, the results presented in Figure 3 elucidate the percentage of Enterobacteriaceae strains that are resistant to the 12 antibiotics evaluated. All Enterobacteriaceae strains evaluated were susceptible to gentamycin, ciprofloxacin, and chloramphenicol, each with 0% resistance. Streptomycin showed a high level of effectiveness against strains tested, except for *Citrobacter freundii,* which showed a 14.3% resistance. *Enterobacter cloacae* and *Proteus mirabilis* showed 12.5% and 66.7% resistance to kanamycin, respectively. Similarly, *Pantoea* spp. and *Providencia rettigeri* showed 50% and 100% resistance to tetracycline, respectively. Cephalothin was only completely effective against *Enterobacter sakazakii*, *Klebsiella Pneumonia*, and *Pseudomonas fluorescence* strains. A total of six Enterobacteriaceae strains, namely, *Enterobacter sakazakii*, *Escherichia hermannii*, *Klebsiella oxytoca*, *Klebsiella Pneumonia*, *Pantoea* spp., and *Pseudomonas fluorescence,* were susceptible to cefpodoxime. Similarly, only *Aeromonas hydrophila*, *Citrobacter youngae*, *Pantoea* spp., *Providencia rettigeri* were susceptible to neomycin. Results from this study are in agreement with the findings of [4], who reported that Enterobacteriaceae isolated from imported fresh produce and US-grown fresh produce were susceptible to gentamycin. While the same study reported that 7.7% and 12.9% of Enterobacteriaceae strains isolated from imported and US-grown fresh produce, respectively, were resistant to kanamycin. Results from our study show that 11.8% of all Enterobacteriaceae strains were resistant to kanamycin. In a similar study, [25] evaluated 10 different commensal and pathogenic genera of Enterobacteriaceae and reported that *Enterobacter cloacae* had the highest occurrence at 42%.

## 4. Conclusions

The findings from this study show that although organically grown vegetables and fruits, such as watermelons, are regarded by consumers as a safer alternative to those grown conventionally due to the absence of pesticide residues and other harmful chemicals, the presence of antibiotic-resistant Enterobacteriaceae strains may pose a food safety concern. Some of the strains isolated from this study have been implicated in food-borne disease outbreaks. The microbial count of Enterobacteriaceae strains was significantly reduced when plastic mulch was applied as a barrier between the soil and fruit surface. The use of plastic mulch was effective in reducing microbial count in watermelons grown at the Tennessee State University farm, and may well be effective on other fruits and vegetables whose edible parts are above ground. Further research should be directed towards investigating the genotypic aspects of antibiotic resistance in the isolates and the correlation between phenotypic and genotypic properties of Enterobacteriaceae strains.

## Figures and Tables

**Figure 1 foods-11-03316-f001:**
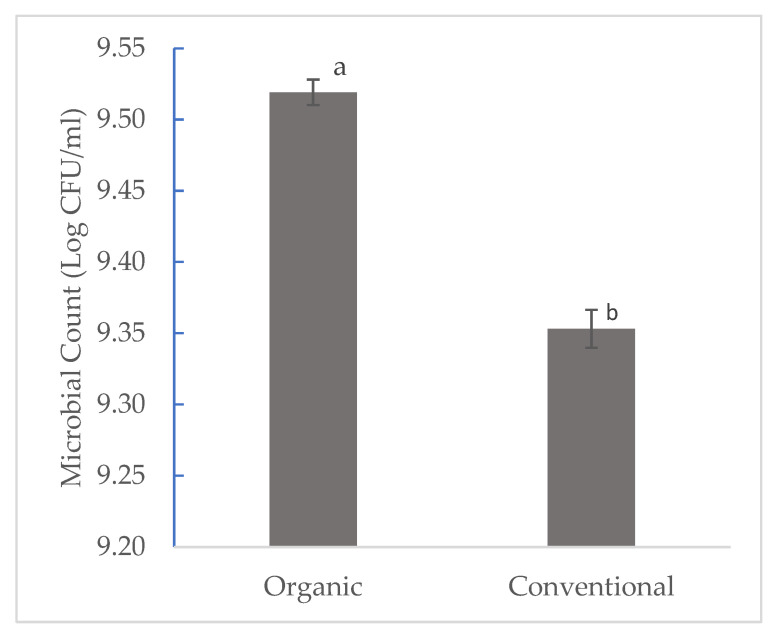
Effects of farming system on Enterobacteriaceae count of watermelon. a–b Mean with the same letters are not significantly different at *p* ˂ 0.05.

**Figure 2 foods-11-03316-f002:**
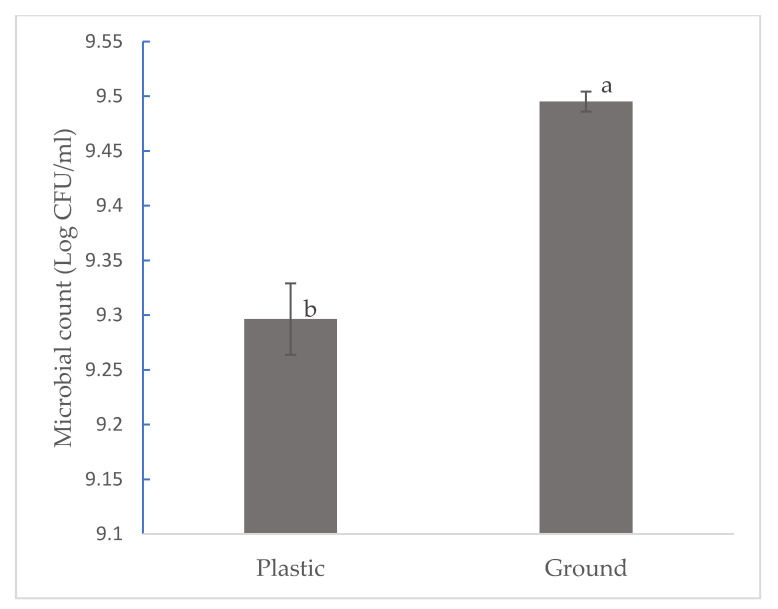
Effects of plastic mulch on Enterobacteriaceae microbial count in organic and conventional watermelon. a–b Mean with the same letters are not significantly different at *p* ˂ 0.05.

**Figure 3 foods-11-03316-f003:**
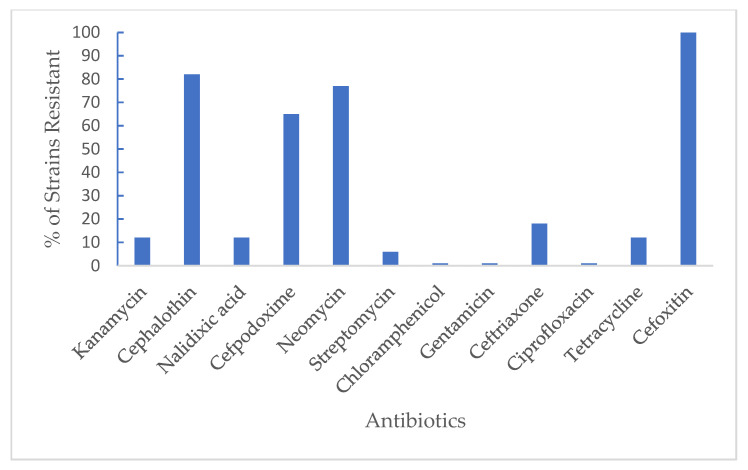
Sensitivity of Enterobacteriaceae strains to antibiotics.

**Table 1 foods-11-03316-t001:** Enterobacteriaceae strains isolated from watermelons in the TSU farm, TN.

Organic Watermelon Field		Conventional Watermelon Field	
Ground	Plastic Mulch	Ground	Plastic Mulch
*Citrobacter freundii*	*Aeromonas hydrophila*	*Enterobacter aerogenes*	*Citrobacter freundii*
*Acinetobacter baumannii*	*Chryseomonas luteola*	*Providential alcalifactiens*	*Enterobacter sakazakii*
*Enterobacter aerogenes*	*Proteus mirabilis*	*Klebsiella oxytoca*	
*Pantoea* spp.	*Citrobacter freundii*	*Citrobacter braakii*	
*Providencia rettigeri*	*Citrobacter youngae*		
*Aeromonas hydrophila*	*Pantoea* spp.		
*Chryseomonas luteola*	*Enterobacter cloacae*		
*Klebsiella oxytoca*	*Klebsiella oxytoca*		
*Enterobacter cloacae*	*Klebsiella Pneumonia*		
*Providential alcalifactiens*			
*Pseudomonas flourescence*			

**Table 2 foods-11-03316-t002:** Sensitivity of Enterobacteriaceae strains isolated from watermelon to antibiotics.

**Strain Type**	**(n)**	**Antimicrobial (s) Resistant Isolates**					
		**C**	**CN**	**CRO**	**CIP**	**TE**	**FOX**
*Acinetobacter baumannii*	3	0(0)	0(0)	2(66)	0(0)	0(0)	3(100)
*Aeromonas hydrophila*	4	0(0)	0(0)	0(0)	0(0)	0(0)	4(100)
*Chryseomonas luteola*	4	0(0)	0(0)	0(0)	0(0)	0(0)	4(100)
*Citrobacter braakii*	2	0(0)	0(0)	0(0)	0(0)	0(0)	2(100)
*Citrobacter freundii*	7	0(0)	0(0)	0(0)	0(0)	0(0)	6(85.7)
*Citrobacter youngae*	2	0(0)	0(0)	0(0)	0(0)	0(0)	2(100)
*Enterobacter aerogenes*	3	0(0)	0(0)	0(0)	0(0)	0(0)	3(100)
*Enterobacter cloacae*	8	0(0)	0(0)	0(0)	0(0)	0(0)	6(75)
*Enterobacter sakazakii*	1	0(0)	0(0)	0(0)	0(0)	0(0)	1(100)
*Escherichia hermannii*	2	0(0)	0(0)	0(0)	0(0)	0(0)	2(100)
*Klebsiella oxytoca*	4	0(0)	0(0)	0(0)	0(0)	0(0)	2(50)
*Klebsiella Pneumonia*	1	0(0)	0(0)	0(0)	0(0)	0(0)	1(100)
*Pantoea spp.*	2	0(0)	0(0)	1(50)	0(0)	1(50)	2(100)
*Proteus mirabilis*	3	0(0)	0(0)	1(33.3)	0(0)	0(0)	3(100)
*Providencia rettigeri*	1	0(0)	0(0)	0(0)	0(0)	1(100)	1(100)
*Providential alcalifactiens*	2	0(0)	0(0)	0(0)	0(0)	0(0)	2(100)
*Pseudomonas flourescence*	1	0(0)	0(0)	0(0)	0(0)	0(0)	1(100)
**Strain Type**	**(n)**	**Antimicrobial (s) Resistant Isolates**					
		**K**	**KF**	**NA**	**CPD**	**N**	**S**
*Acinetobacter baumannii*	3	0(0)	3(100)	0(0)	2(66.7)	1(33)	0(0)
*Aeromonas hydrophila*	4	0(0)	3(75)	1(25)	1(25)	0(0)	0(0)
*Chryseomonas luteola*	4	0(0)	2(50)	0(0)	2(50)	2(50)	0(0)
*Citrobacter braakii*	2	0(0)	1(50)	0(0)	2(100)	1(50)	0(0)
*Citrobacter freundii*	7	0(0)	3(42.8)	0(0)	7(100)	4(57.1)	1(14.3)
*Citrobacter youngae*	2	0(0)	1(50)	0(0)	2(100)	0(0)	0(0)
*Enterobacter aerogenes*	3	0(0)	3(100)	0(0)	2(66.7)	2(66.7)	0(0)
*Enterobacter cloacae*	8	1(12.5)	6(75)	0(0)	4(50)	3(37.5)	0(0)
*Enterobacter sakazakii*	1	0(0)	0(0)	0(0)	0(0)	1(100)	0(0)
*Escherichia hermannii*	2	0(0)	2(100)	0(0)	0(0)	1(50)	0(0)
*Klebsiella oxytoca*	4	0(0)	2(50)	0(0)	0(0)	3(75)	0(0)
*Klebsiella Pneumonia*	1	0(0)	0(0)	0(0)	0(0)	1(100)	0(0)
*Pantoea spp.*	2	0(0)	2(100)	0(0)	0(0)	0(0)	0(0)
*Proteus mirabilis*	3	2(66.7)	1(33.3)	0(0)	2(66.7)	1(33.3)	0(0)
*Providencia rettigeri*	1	0(0)	1(100)	0(0)	1(100)	0(0)	0(0)
*Providential alcalifactiens*	2	0(0)	2(100)	1(50)	1(50)	2(100)	0(0)
*Pseudomonas flourescence*	1	0(0)	0(0)	0(0)	0(0)	1(100)	0(0)

Values in parenthesis show the % resistance of Enterobacteriaceae strain to the selected antibiotics. chloramphenicol (C), gentamicin (CN), ceftriaxone (CRO), ciprofloxacin (CIP), tetracycline (TE), cefoxitin (FOX), kanamycin (K), cephalothin (KF), nalidixic acid (NA), cefpodoxime (CPD), neomycin (N), and streptomycin (S).

## Data Availability

Data is contained within the article.

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
