# Peer review of "Incidence of Drug-Resistant Enterobacteriaceae Strains in Organic and Conventional Watermelons Grown in Tennessee"

_foods, 2022, doi:10.3390/foods11213316_

Round 1

Reviewer 1 Report (Previous Reviewer 2)

I thank the authors for revising the manuscript. The manuscript has been improved. There are some minor issues that need to be addressed.

There was no description of Fig 1 in the main text of the manuscript. 

The caption of Fig 1 should be better described. What is the clear zone and what is the white round pad?

Tick marks should be added to the y-axis in figure 2a and 2b and figure 3.

Author Response

Comments and Suggestions for Authors

I thank the authors for revising the manuscript. The manuscript has been improved. There are some minor issues that need to be addressed.

There was no description of Fig 1 in the main text of the manuscript. 

The caption of Fig 1 should be better described. What is the clear zone and what is the white round pad?

Response:

After considering your comments and the comments of Reviewer #3 to delete Figure 1, the authors have decided to delete Figure 1.

Tick marks should be added to the y-axis in figure 2a and 2b and figure 3.

Response:

Tick marks have been added to the y-axis in Figures 2a, 2b, and figure 3.

Reviewer 2 Report (New Reviewer)

Dear authors,

The manuscript entitled “Incidence of Drug-Resistant Enterobacteriaceae Strains in Organic and Conventional Watermelons Grown in the Tennessee State University Farm” is appropriately written and structured by Akaeze et al., strongly appropriate English with a clear structure. They investigated antimicrobial resistance in Enterobacteriaceae isolated from organic and conventional watermelon fruits. They isolated 17 strains from watermelon samples. They found that the isolates were susceptible to Gentamycin, Ciprofloxacin, and Chloram phenicol but were resistant to cefoxitin. The results are very interesting; however, as a full paper for publishing in a high-ranked and prestigious journal (the Foods journal), the data and results are so limited. I strongly recommend developing the research and investigating genotypic aspects of antibiotic resistance in the isolates. You can use NGS or conventional PCR methods for the detection of antibiotic resistance genes in the isolates and address the correlation between the phenotypic and genotypic characteristics. After developing the research, this manuscript should be re-considered for critical evaluation. Consequently, it is recommended to

1.      Investigate and detect antibiotic resistance genes in isolates (by using NGS or PCR techniques according to the budget of the research)

2.      Implement statistical procedures to find out the correlation between phenotypic and genotypic properties. 

Author Response

Comments and Suggestions for Authors

Dear authors,

The manuscript entitled “Incidence of Drug-Resistant Enterobacteriaceae Strains in Organic and Conventional Watermelons Grown in the Tennessee State University Farm” is appropriately written and structured by Akaeze et al., strongly appropriate English with a clear structure. They investigated antimicrobial resistance in Enterobacteriaceae isolated from organic and conventional watermelon fruits. They isolated 17 strains from watermelon samples. They found that the isolates were susceptible to Gentamycin, Ciprofloxacin, and Chloram phenicol but were resistant to cefoxitin. The results are very interesting; however, as a full paper for publishing in a high-ranked and prestigious journal (the Foods journal), the data and results are so limited. I strongly recommend developing the research and investigating genotypic aspects of antibiotic resistance in the isolates. You can use NGS or conventional PCR methods for the detection of antibiotic resistance genes in the isolates and address the correlation between the phenotypic and genotypic characteristics. After developing the research, this manuscript should be re-considered for critical evaluation. Consequently, it is recommended to

  1. 1.      Investigate and detect antibiotic resistance genes in isolates (by using NGS or PCR techniques according to the budget of the research)
  2. Implement statistical procedures to find out the correlation between phenotypic and genotypic properties. 

Response:

Thank you very much for your thorough critique. Results from our study, in addition to isolating 17 Enterobacteriaceae strains and their sensitivity to 12 antibiotics also contain results showing that watermelon fruits that were directly on the ground had significantly higher microbial content when compared to those on plastic mulch. Your suggestions although very helpful are outside the scope of the research conducted but have been included in the conclusion of the manuscript for future research

Reviewer 3 Report (New Reviewer)

This paper focused on the prevalence and antibiotic susceptibility of Enterobacteriaceae isolated from watermelons grown on the Tennessee State University farm. The study has a certain application value and has a certain guiding significance for food safety. However, only phenotypic observations have been presented, but the essential reason is not supported by data. In addition, the experimental design is relatively simple, and the research is not thorough enough. Detailed comments are as follows:

(1) The samples were restricted to a farm in Tennessee State University. It is unknown whether the findings hold true for other farms.

(2) How many samples were taken for the isolation of Enterobacteriaceae?

(3) Figure 1 does not make sense and it should be deleted.

(4) Line 107: How could you assure the accuracy of the antimicrobial susceptibility tests without reference strains such as E. coli ATCC 25922?

(5) Data difference between the two groups was minor (less than 0.2 log units). In this sense, the related conclusions were not solid.

Author Response

This paper focused on the prevalence and antibiotic susceptibility of Enterobacteriaceae isolated from watermelons grown on the Tennessee State University farm. The study has a certain application value and has a certain guiding significance for food safety. However, only phenotypic observations have been presented, but the essential reason is not supported by data. In addition, the experimental design is relatively simple, and the research is not thorough enough. Detailed comments are as follows:

The samples were restricted to a farm in Tennessee State University. It is unknown whether the findings hold true for other farms.

Response:

As shown by the title and as acknowledged by Reviewer #1 as being appropriate, the study seeks to determine the presence or absence of drug-resistant Enterobacteriaceae strains only in the Tennessee State University Farm.

How many samples were taken for the isolation of Enterobacteriaceae?

Response:

As shown in lines 85-87, 9 fruits were selected each from fruits laying on plastic mulch and from fruits laying on the soil of both organic and conventional plots respectively. In the laboratory, each selected watermelon was thoroughly scrubbed in 20 ml of peptone water to dislodge the microbes present on fruit surfaces.  Next, seven-fold serial dilutions were prepared in Tryptic Soy Broth (TSB) in four replicates for microbial analysis and quantification. Microbial isolation and purification were done by the spread plate technique on violet red bile agar by picking five colonies (with characteristic red to dark purple with red-purple halos)  per replicate and incubating at 37O C for 24 h.

Figure 1 does not make sense and it should be deleted.

Response:

Figure 1 has been deleted.

Line 107: How could you assure the accuracy of the antimicrobial susceptibility tests without reference strains such as E. coli ATCC 25922?

Response:

The results were interpreted as susceptible, intermediate, and resistant based on the Clinical and Laboratory Standards Institute recommendations. Escherichia coli ATCC 25922 and Staphylococcus aureus ATCC 25923 were used as control strains. Reference standard bacterial strains were verified simultaneously with controls.

Data difference between the two groups was minor (less than 0.2 log units). In this sense, the related conclusions were not solid.

Response: Data analysis was done by ANOVA using the SAS statistical software (version 9.4, SAS Institute). Results for mean separation showed that values were significant. Please find attached exerts from the SAS output.

Round 2

Reviewer 2 Report (New Reviewer)

All revisions have been addressed. 

Author Response

Thank you very much for your thorough critique. Results from our study, in addition to isolating 17 Enterobacteriaceae strains and their sensitivity to 12 antibiotics also contain results showing that watermelon fruits that were directly on the ground had significantly higher microbial content when compared to those on plastic mulch. Your suggestions although very helpful are outside the scope of the research conducted but have been included in the conclusion of the manuscript for future research.

Reviewer 3 Report (New Reviewer)

Although the authors have addressed some of the comments, there are still some revisions needed for this manuscript. Detailed comments are as below.

(1) I strongly recommend changing the article type to “Communication”. The experimental design is simple, and the research is not thorough enough. The presented data are not enough for a Research Article.

(2) Please add “Conventional watermelon” to the Keywords.

(3) Figures are of low quality. The Y-axis was missing in Figure 2a and Figure 2b. These figures should be reproduced.

(4) Since Figure 1 has been deleted, other figures should be renumbered. “Figure 2a” and “Figure 2b” should be changed to “Figure 1” and “Figure 2” (lines 185-186, 189-190).

(5) Line 189-190: …in organic and conventional watermelons.

(6) Table 2a and 2b should be combined into one Table.

(7) Figure 3 should be remade. Please put the antibiotic names in the figure captions, instead of the figure itself.

(8) Lines 21-22: Why did you only select 9 fruits? More fruits are needed for a robust microbial analysis.

Author Response

Comments and Suggestions for Authors:

Although the authors have addressed some of the comments, there are still some revisions needed for this manuscript. Detailed comments are as below.

(1) I strongly recommend changing the article type to “Communication”. The experimental design is simple, and the research is not thorough enough. The presented data are not enough for a Research Article.

Response: Comment addressed in the revised manuscript.

(2) Please add “Conventional watermelon” to" the Keywords.

Response:

“Conventional” has been added to keywords

(3) Figures are of low quality. The Y-axis was missing in Figure 2a and Figure 2b. These figures should be reproduced.

Response:

Figures 2a and 2b are not missing the Y-axis. #Reviewer 1 requested for tick marks to be included to the Y-axis. That is why the figure appears like that.

(4) Since Figure 1 has been deleted, other figures should be renumbered. “Figure 2a” and “Figure 2b” should be changed to “Figure 1” and “Figure 2” (lines 185-186, 189-190).

Response:

Figures has been renamed accordingly (Figure1, Figure 2 and Figure 3)

(5) Line 189-190: …in organic and conventional watermelons.

Response:

The Title of Figure 2 has been recast as follows “Effect of plastic mulch on Enterobacteriaceae microbial count in organic and conventional watermelon”

(6) Table 2a and 2b should be combined into one Table.

Response:

Table 2a and Table 2b are the same table. It was split in two due to space. The title has been changed to Table 2 and Table 2 cont’d.

(7) Figure 3 should be remade. Please put the antibiotic names in the figure captions, instead of the figure itself.

Response:

The Figure 3 has been remade accordingly

(8) Lines 21-22: Why did you only select 9 fruits? More fruits are needed for a robust microbial analysis.

Response:

As shown in lines 86-88, “At harvest, 9 fruits were selected each from fruits laying on plastic mulch and from fruits laying on the soil of both organic and conventional plots respectively. That is a total of 36 watermelons and not 9 watermelons. Lines 21-22 has been recast to prevent further misunderstanding.

This manuscript is a resubmission of an earlier submission. The following is a list of the peer review reports and author responses from that submission.

Round 1

Reviewer 1 Report

Major comments:

1. Please list the fertilizers used by the two farming methods. Organic farming may use animal manure, while conventional farming uses chemical fertilizers. This may be the reason for the high microbial pollution caused by organic fertilizer. Add to these discussions.

2. The collection of experimental samples is only carried out in a certain farm, and the sample size is only 9 fruits. The regional selection of the samples was not universal and the sample size was too small, so the study could only account for the contamination of antibiotic-resistant Enterobacteriaceae on a particular farm.

3. Drug-resistant Enterobacteriaceae were isolated in both ordinary farms and organic farms, but in sentences 26-28, only antibiotic-resistant Enterobacteriaceae strains were found in organic watermelons, which contradicts the title.

4. The processing and layout of the figure is not beautiful enough.

5. The purpose of the experiment is not clear enough. Was it to compare the proportion of contamination by antibiotic-resistant strains from organic and conventional watermelon? If so, the results section should show the resistant isolates isolated from organic and conventional watermelon, rather than simply stating that 17 Enterobacteriaceae isolates were isolated in all samples.

General comments:

1. There is no figure 1 in the article.

2. The figures do not have an ordinate axis.

3. Do not need quotation marks in the title of the article in the References.

Line 68: Change "antibiotic" to "antibiotics".

Author Response

Major comments:

  1. Please list the fertilizers used by the two farming methods. Organic farming may use animal manure, while conventional farming uses chemical fertilizers. This may be the reason for the high microbial pollution caused by organic fertilizer. Add to these discussions.

Response: The fertilizers used has been included (lines 84-86). Organic manure is not used on the farm the study was conducted. Lines 177-178 already explained the possible reason for a higher microbial count in organic fields

  1. The collection of experimental samples is only carried out in a certain farm, and the sample size is only 9 fruits. The regional selection of the samples was not universal and the sample size was too small, so the study could only account for the contamination of antibiotic-resistant Enterobacteriaceae on a particular farm.

Response: The experiment is a part of a much bigger experiment that was conducted in one location. The title of the MS has been reviewed to indicate the location of the experiment (Antimicrobial drug-resistant Enterobacteriaceae in organic and conventional watermelon (Citrullus lanatus) fields in Nashville, TN)

  1. Drug-resistant Enterobacteriaceae were isolated in both ordinary farms and organic farms, but in sentences 26-28, only antibiotic-resistant Enterobacteriaceae strains were found in organic watermelons, which contradicts the title.

Response: The sentences in line 26-28 has been corrected to avoid misunderstanding. Antibiotic-resistant strains were found in both organic and conventional watermelons.

  1. The processing and layout of the figure is not beautiful enough.

Response: This has been corrected.

  1. The purpose of the experiment is not clear enough. Was it to compare the proportion of contamination by antibiotic-resistant strains from organic and conventional watermelon? If so, the results section should show the resistant isolates isolated from organic and conventional watermelon, rather than simply stating that 17 Enterobacteriaceae isolates were isolated in all samples.

Response: The thrust of this study is not a comparison of the proportion of contamination by antibiotic-resistant isolates isolated from organic and conventional watermelon. It is to determine the presence or absence of antibiotic resistance in organic and/or conventional watermelon. This is important because it is a general opinion that any organic product is safe. This study has shown that the high microbial count and presence of antibiotic-resistant Enterobacteriaceae strains may raise safety concerns even in organic products

General comments:

  1. There is no figure 1 in the article.

Response: It has been corrected

  1. The figures do not have an ordinate axis.

Response: Figures have been updated

  1. Do not need quotation marks in the title of the article in the References.

Line 68: Change "antibiotic" to "antibiotics".

Response: "Antibiotic" has been changed to "antibiotics" in line 68. Also, the quotation marks in the title of articles in the references have been removed

Reviewer 2 Report

The manuscript presents a study on the diversity of microorganisms on organic and conventional watermelons as well as the antibiotic resistance of the Enterobacteriaceae on two different watermelons. Results showed that organic watermelon had more diversity of microbes and antibiotic resistance was discussed. However, there are some issues that need to be addressed.

Material and methods section: I suggest dividing the whole section into sub-sections such as materials, bacterial culture, strains identification, etc.

There is no Figure 1 in the manuscript. It starts with Figure 2.

There is no description of Figure 2 in the main text.

Figure 3. There is no error bar in the figure. How many replicates were performed for the study? The meaning of “a” and “b” in figure 3a should be described in the caption.

There is also no error bar in figure 4

Figure 3b needs more clarification. What analysis was done for the figure? Which one is organic or conventional watermelon is not clear.

The quality of the figures in the manuscript can be improved.

The method of statistical analysis should be discussed in the manuscript.

The manuscript lacks a deep discussion about the results. It discussed the results in this study were similar to the previous studies. Then what is the novelty and significance of this study? The novelty and significance should be emphasized.

Also for the conclusion, the results were described again without a comprehensive conclusion on the significance of this study and future direction.

Author Response

Review Report 1:

Comments and Suggestions for Authors

The manuscript presents a study on the diversity of microorganisms on organic and conventional watermelons as well as the antibiotic resistance of the Enterobacteriaceae on two different watermelons. Results showed that organic watermelon had more diversity of microbes and antibiotic resistance was discussed. However, there are some issues that need to be addressed.

Material and methods section: I suggest dividing the whole section into sub-sections such as materials, bacterial culture, strains identification, etc.

Response: The Material and methods section has been divided in sections (Experimental location and design, microbial isolation, microbial identification, antimicrobial susceptibility test and statistical analysis)

There is no Figure 1 in the manuscript. It starts with Figure 2.

Response: This has been corrected

There is no description of Figure 2 in the main text.

Response: This has been corrected (line 111)

Figure 3. There is no error bar in the figure. How many replicates were performed for the study? The meaning of “a” and “b” in figure 3a should be described in the caption.

Response: Error bars has been included to the figure. Field experimental layout was replicated three times while laboratory experiments were replicated 4 times (lines 81 and 90). Also the meaning of “a” and “b” has been included.

There is also no error bar in figure 4

Response: The data presented in this figure is in percentage, hence no error bar.

Figure 3b needs more clarification. What analysis was done for the figure? Which one is organic or conventional watermelon is not clear.

Response: The title of the figure has been clarified. As described in the “statistical analysis” section of the material and method, “All plate count data were converted to log CFU per ml values before statistical analysis. Data were analyzed using Analysis of variance (ANOVA) and means were separated by LSD using the SAS statistical software (version 9.4, SAS Institute).

The quality of the figures in the manuscript can be improved.

Response: This has been corrected

The method of statistical analysis should be discussed in the manuscript.

Response: This has been included in the “statistical analysis” section of the material and methods

The manuscript lacks a deep discussion about the results. It discussed the results in this study were similar to the previous studies. Then what is the novelty and significance of this study? The novelty and significance should be emphasized.

Also for the conclusion, the results were described again without a comprehensive conclusion on the significance of this study and future direction.

Response: while it is true that there are many literature on farming system on microbial dynamics of many vegetable crops, there is a dearth of literature on the effect of farming systems on watermelon in particular. This is where this study focus on. Furthermore, unlike many other studies, this study raises food safety concerns on organic watermelon due to the presence of antimicrobial resistant Enterobacteriaceae strains on watermelon.

The discussion and conclusion has been adequately revised to convey this information.
